# Proprietary Varieties' Influence on Economics and Competitiveness in Land Use within the Hop Industry

**Douglas MacKinnon** [1] **and Martin Pavlovič** [1,2,*]

1   Department of Agriculture Economics and Rural Development, Faculty of Agriculture and Life Sciences, University of Maribor, 3211 Hoče, Slovenia
2   Slovenian Institute of Hop Research and Brewing, 3310 Žalec, Slovenia
*   Correspondence: martin.pavlovic@um.si

**Abstract:** To evaluate changes to hop industry concentration and competitiveness the Herfindahl-Hirschman Index (HHI) was used. The ownership of hop proprietary varieties, their acreage and production were compared with public varieties. Market share for each proprietary hop variety acreage and production was calculated between 2000 and 2020. The quantity of land under centralized control in the U.S. hop industry due to increased proprietary variety acreage between 2000 and 2020 was quantified. Assuming tacit collusion between the participants in the oligopoly, the HHI enabled us to quantify the portion of land under oligopoly control. The HHI analysis of hop acreage and hop production demonstrated that market concentration rose rapidly between the years 2010 (0.0376 and 0.0729) and 2020 (0.4927 and 0.5394). This resulted in decreasing business competitiveness within the market during this period caused primarily by rapid consolidation of ownership during increased proprietary variety acreage and production increases. Calculations revealed that in 2016 a tipping point had been reached concerning market concentration, which resulted in higher sustained season average prices of hops—a key raw material in brewing.

**Keywords:** hop industry; varieties; market concentration; intellectual property; prices

## 1. Introduction

Hops, along with malt and water, are the basic raw materials used for beer production. The basic role of hops is to provide beer with a pleasantly bitter taste and a hoppy aroma [1–3]. Between 2000 and 2020 the proportion of privately owned patented U.S. hop varieties increased. Other countries experienced increases in patented varieties, but these were owned by national associations. In the United States during that time, one variety development company, the Hop Breeding Company, grew to the point where its varieties enjoyed significant market share. The agglomeration of hop farms in Washington, Oregon, and Idaho (i.e., the Pacific Northwest (PNW)) facilitates the exchange of information by reducing monitoring costs thereby increasing market transparency among participants [4]. The increase in proprietary variety acreage and production has a causal effect on hop prices [5]. Tacit collusion results from competitors independently realizing their collective best interests to adjust prices or quantities [6]. The exchange of production-related information including anticipated yields and current prices among farmers may lead to a similar outcome [7].

Since 1913, the United States Department of Agriculture (USDA) has collected and published statistical data regarding the U.S. hop industry. The publication of intellectual property (IP) necessitates the use of symbols for registered trademarks, unregistered trademarks, and copyright, (i.e., "®", "™" and "©") respectively. The USDA complies with these requirements. Proprietary variety ownership is publicly available through the U.S. Patent and Trademark Office (USPTO). The introduction of proprietary varieties, therefore, enabled the calculation of hop market share by acreage and production for the first

time. Market share regarding sales of these varieties to brewers remained unavailable but was not important for calculating influence within the U.S. hop industry.

We used the Herfindahl-Hirschman Index (HHI) to measure changes in hop industry competitiveness by way of measuring market concentration. A similar methodological approach was used to measure market concentration in the airline industry [8]. According to the 2020 U.S. Federal Register, HHI was used to evaluate the acquisition of the Craft Brew Alliance, Inc. (CBA) by Anheuser-Busch InBev SA/NV ("ABI") and Anheuser-Busch Companies, LLC ("AB Companies"). The results of such analyses can provide insights into industry behavior. Markets with relatively high HHI values, market share inequality, and the presence of major firms are imperfectly competitive. Under such circumstances, market imperfections are vulnerable to exploitation [9].

The presence of IP introduced constraints into the market where none had previously existed and affected farmer planting decisions on that acreage. More constrained varieties were planted at a faster pace than those that were unconstrained [10]. Changes in market concentration and price-cost margins can be used to determine the direction of competitiveness [11]. The greater degree of specificity, control, and profit incentivized private hop breeding companies to invest further in the development of new intellectual property. Their owners are incentivized by the ability to protect and enforce their rights [12]. Patent law also enabled IP owners to determine production and distribution via licensing agreements.

In 2021, Germany and the U.S. produced 32.91 and 40.87 percent of global hop acreage respectively. These are the two largest hop-producing regions, but each operates under different business models. In 2020, patented and trademarked varieties in the United States represented 70.19% of PNW acreage and 73.44% of PNW production. Between 2009 and 2019, the annual farmgate value of American hops increased by 282% [13]. According to the country reports of the International Hop Growers' Convention (IHGC) between 2009 and 2019, 70 hop farmers in the PNW received approximately $4.7 billion in farm revenue, $2.88 billion more than the $1.87 billion the 1087 German hop farmers received during the same period [14].

Between 1998 and 2020, USDA data recorded U.S. proprietary variety acreage and production soaring from zero to over 70 percent. Publicly available information regarding proprietary variety ownership enabled us to calculate the U.S. hop market share for the first time in history. One variety development company, the Hop Breeding Company (HBC), owned the varieties responsible for over 50 percent of U.S. acreage and production by 2020 [13].

The objective of the study was to evaluate changes to the hop industry area and production concentration and competitiveness with respect to the changes in proprietary varieties of hops relative to public varieties. Comi [15] refers to farmers who euphemistically described this process as "decommodification" implying only positive added value for the good of all. Using the Herfindahl-Hirschman Index (HHI), the concentration of acreage-producing proprietary varieties between 2000 and 2020 under the control of a cartel-like structure with strict production and sales licensing agreements was quantified. Proprietary variety owners used their IP to create competitive advantages for their companies and those farmers allied with them. Consequently, they denied their competitors and farmers they did not favor primary access to their IP. This research analyzed publicly available industry data to determine the market effects resulting from the increased use of branded proprietary varieties by the craft brewing industry during this time and compared it with other periods possessing unique characteristics dating back to 1948.

## 2. Materials and Methods

### 2.1. Proprietary Hop Variety Supply and Market Share

The United States Department of Agriculture lists each branded proprietary variety together with their respective intellectual property symbols in their publications [16]. The details of patents and trademarks are public information. By tracking the ownership of these varieties in patent and trademark records with the U.S. Patent and Trademark Office

(USPTO) and through the Google Patent Search website, we discerned the influence of individuals and entities over proprietary varieties. Data reported by the USDA included season average prices (SAP), inventory levels, production, and acreage. Wright and Williams [17] suggest that when supply is elastic and demand inelastic (as is the case with hops), the accumulation of stocks is typically damped by a compensating production response. The hop industry suffers from something called the Delayed Surplus Response (DSR). Production is highly elastic when prices and demand increase, but there is a delay of several years when prices and demand decrease. This results in surplus production that negatively affects global prices for hops through recurring boom-and-bust cycles [18]. Data published by the USDA enabled us to calculate the accumulation of aggregate stock levels and the annual market share of acreage by variety. We restricted our research to USDA National Hop Report (NHR) data between 1998 and 2022. That represented the period during which branded proprietary varieties were first reported by the USDA and included the most recently available industry data at the time of our calculations.

The companies that developed proprietary hop varieties own and license the production of multiple proprietary varieties to growers (for production) and sales and distribution of those varieties to merchants in their supply chain, thereby facilitating the management of production and distribution. We calculated the percentage for each proprietary variety produced within the Pacific Northwest (PNW) by the total acreage for the PNW i.e., the total market share. We calculated the market share for each entity owning IP listed by the USDA NASS in the USDA National Hop Report (NHR) by grouping those with common ownership of patented and trademarked products [16].

We expanded the variety-specific acreage market share calculations to group together those varieties that share common ownership to get a better picture of the influence of the five largest variety development companies. One company, the Hop Breeding Company, LLC (HBC) enjoyed increased influence within the industry as its proprietary varieties increased to occupy 51% of the acreage in the PNW. According to the company's website (www.hopbreeding.com, accessed on 18 July 2022), it is a joint venture between John I. Haas, Inc., a hop merchant company, and Yakima Chief Ranches, L.L.C., a company owned by the Smith, Carpenter, and Perrault families. These three families are also shareholders of Yakima Chief Hops, Inc., a hop merchant company. This complicated ownership structure effectively created a duopoly through which the proprietary varieties of the HBC were processed and distributed. Production was handled first through the farming resources to which each of the merchant companies had access. In the case of John I. Haas, Inc., that included the company farm and entities such as Roy Farms Inc., who touts on their website (http://royfarms.com/hops/roy-farms-citra/, accessed on 15 November 2022) their production and direct sales of one popular proprietary HBC variety, Citra®. In the case of Yakima Chief Hops Inc., those resources belonged to their farm owners. According to an article in the beer industry news outlet, Brewbound [18], that number expanded in 2019 from 11 to 15 farms. These varieties created a competitive advantage for the shareholders of the HBC and the other companies with whom it shared common ownership. How the owners of the HBC managed acreage between the two merchant companies to maintain equitable market share remains unknown. Proprietary varieties were distributed worldwide via licensing agreements with select merchants. The influence over such substantial acreage afforded the individuals involved with the HBC a disproportionate amount of influence in the industry. Their patents enabled them to decide via licensing agreements who would produce and sell their varieties. The MacKinnon Report, a hop market report published on Substack.com (https://mackinnonreport.substack.com, accessed on 26 January 2023) detailed that in 2023 the patent owners must reduce proprietary variety acreage by at least 8328 acres (3371 ha.) in response to a massive surplus that began in 2016. Those decisions have the power to create inefficiencies for some farms. How the decisions to reduce acreage will be coordinated by the two companies and which farmers will be told to reduce acreage remains to be seen. Some will be less efficient producers and not be able to compete in the future market.

Data demonstrated that the difference between depletion rates (i.e., the quantity of hops shipped from U.S. warehouses) and the total available supply increased by 54 million pounds between 2016 and 2022 (Figure 1). What we could not ascertain, however, was the degree to which inventories managed by the same companies were growing in warehouses located outside the purview of the USDA NASS surveys (i.e., every country other than the U.S.). Other countries do not publish similar data, which limited the completeness of this study.

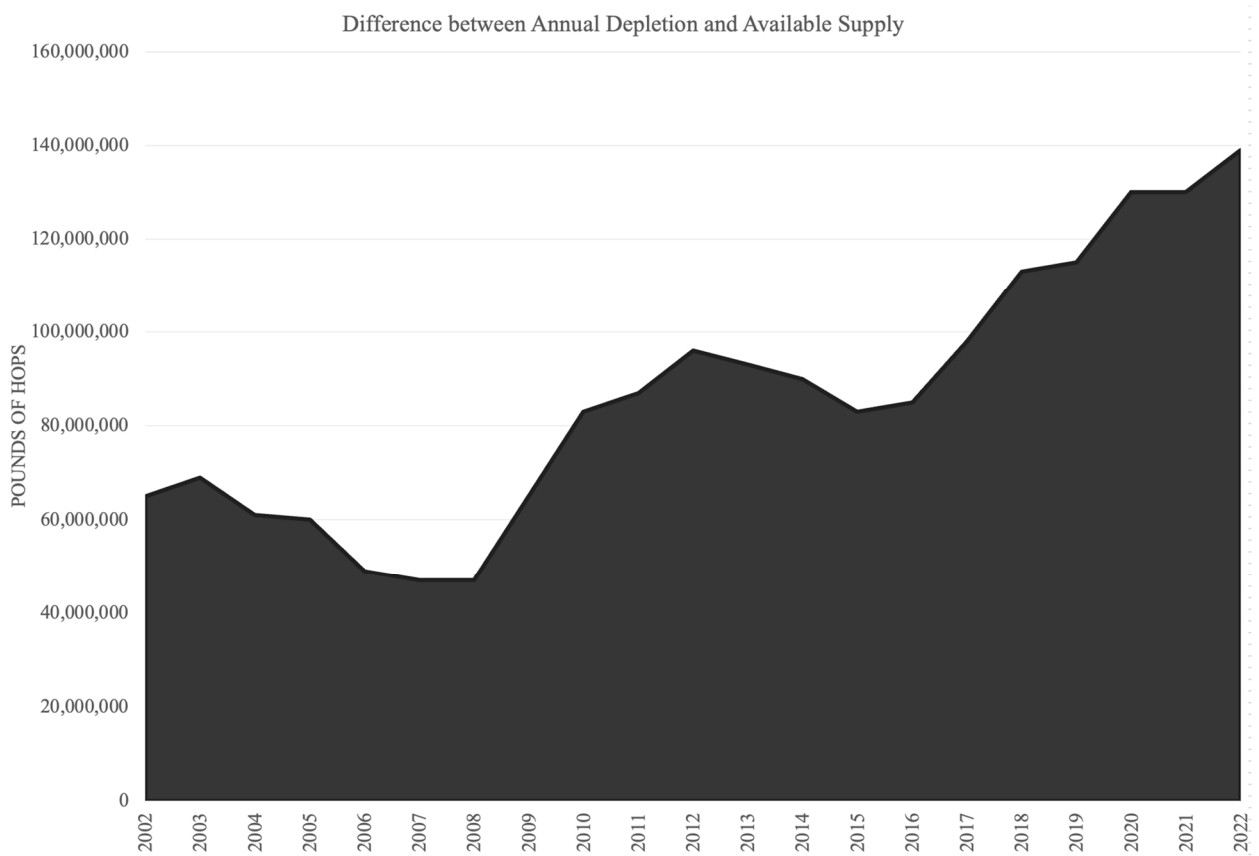

**Figure 1.** Annual Depletion (Sept n–Sept n-1) Relative to Total Available Supply (Crop (n-1) and Inventory (n-1)) (own study based on [16]).

### 2.2. Calculating HHI

We used the Herfindahl-Hirschman Index (HHI) to evaluate changes to hop industry concentration and competitiveness with respect to the changes in proprietary varieties of hops relative to public varieties. A significant portion of PNW acreage appears in two aggregate categories called "other" and "experimental". USDA data show that in 2022 7.12% of total acreage was categorized as "other" or "experimental". The categories are used to report acreage and production for varieties that do not meet the three-independent-grower threshold set by the USDA. Based on historical data, we believe at least half of this acreage was proprietary.

The HHI is a method used also by the United States Department of Justice (USDOJ) to measure market concentration during mergers or acquisitions, to evaluate one competitor's position relative to another, and to uncover potentially anti-competitive practices. The HHI values of zero to 0.1500 mean a low market concentration. Values of 0.1500 to 0.2500 are considered a moderate concentration. Values of 0.2500 and above count as high concentration. The HHI value will be low when market shares among participants are equal. The value will be high when one firm has a disproportionate share of the market [19]. The value of the HHI decreases as the number of firms in the market increases. Market

concentration is inversely proportional to competitiveness [20]. The HHI is responsive to the asymmetry of market shares and is used to evaluate changes in the competitiveness within a single industry over time or to compare one industry to another [19]. In our research, we adopted this method for the first time in the hop industry for the measurement of its market concentration.

The HHI Formula

$$HHI = S1^2 + S2^2 + S3^2 + \ldots Sn^2 \tag{1}$$

where:

n refers to the number of varieties in the market;
S refers to the percent market share for a variety.

## 3. Results and Discussion

### 3.1. Acreage and Production Linked to Proprietary Hop Varieties

Calculating the Herfindahl-Hirschman Index (HHI) of the U.S. hop industry based on the market share of hop sales to brewers was a hopeless endeavor as information regarding market share based on sales of hops by merchants to breweries was proprietary information and never shared. We discovered an alternative method for measuring market share. The USDA NASS restrictions related to the reporting of proprietary U.S. acreage and production (i.e., that three or more independent producers must list acreage or production for the corresponding statistics to be reported in aggregate form) to meet the needs of this research.

Acreage, and the infrastructure necessary to harvest that acreage, was the scarcest and most valuable commodity in the hop industry in 2020, not the hops themselves. Acreage was the asset for which there was the greatest competition. The primary method for harvesting hops was via fixed picking machine facilities. Mobile combines exist that harvest cones from the vines in the field. Combines, when they were used, operated in conjunction with the more traditional fixed-picking facilities that could process at least 600 acres (242 hectares) of hops in a season. Combines returned cones harvested in the field to the picking facility to separate leaves, stems, and foreign material from the cones themselves through the picking facility's recleaning equipment. Due to the time-sensitive nature of the harvest, high ambient air temperatures, which could reach over 100 degrees Fahrenheit (37.78 °C ) in Washington and Idaho states during harvest, hop growers sought to grow hops on land that was in close proximity (not more than 10–15 min driving time) to their fixed picking facilities to reduce the incidents of hops dehydrating prior to going through the picking machine, which increases cone shatter and reduces quality.

Five companies comprised approximately 70 percent of proprietary U.S. hop acreage and production in the Pacific Northwest in 2022 (Table 1). These variety development companies can license hop merchant companies to sell their varieties. They can license hop farms to produce their varieties. In some cases, the variety of development company ownership and the licensed merchants and farms shared common ownership. Licenses extended beyond companies in which they shared ownership. Previously independent farms were transformed into contract growers. The decision-makers for the five largest variety development companies, therefore, enjoyed a disproportionate influence in the industry and upon the market. The acreage on which a company's proprietary varieties were produced represented the market share of influence of the owners of each variety development company. The market share of influence represented a new and significant measurement possible within the industry all made possible by the growing demand for and reporting of proprietary varieties of hops.

**Table 1.** Five largest U.S. hop variety development companies and the market share of U.S. acreage and production of their proprietary varieties in 2022.

|  | **Variety Development Company** | **Market Share of U.S. Total Acreage in 2022** | **Market Share of U.S. Total Production in 2022** |
|---|---|---|---|
| **1** | Hop Breeding Company (HBC) | 49.05% | 49.12% |
| **2** | HopSteiner | 7.71% | 10.78% |
| **3** | Association for the Development of Hop Agronomy (ADHA) | 3.27% | 3.35% |
| **4** | Virgil Gamache Farms (VGF) | 3.20% | 2.88% |
| **5** | CLS Farms | 2.06% | 2.02% |
|  | TOTAL | 65.30% | 68.15% |

Calculating the market share for each ownership group based on their ownership of proprietary hop varieties enabled the calculation of the market share of influence over the scarcest resource in the hop industry, acreage. Branded proprietary varieties are products that enjoy monopoly control by their very nature as patented and trademarked products. Seventy percent of the acreage, therefore, was governed by the decision-makers of five entities. Public varieties, in contrast, are available for any grower to produce.

We calculated the market share for each proprietary variety production and acreage relative to total U.S. acreage for the years 2000 through 2020. During this time, market concentration moved from low to high according to the standards set by the U.S. Department of Justice when evaluating mergers and acquisitions between competitors.

Using the HHI market share data by variety, we calculated the market share for all proprietary varieties collectively as the U.S. hop industry resembles what is referred to as a complex monopoly in the U.K. [21]. We calculated the increase in market concentration between 2000 and 2020 of publicly reported U.S. proprietary hop varieties. The increasing HHI values between 2000 and 2020 demonstrated the changes in the degree of competitiveness in the industry (Table 2).

**Table 2.** HHI Values for U.S. Total Proprietary Varieties by Acreage and Production 2000–2020.

| **Crop Year** | **HHI Values for Proprietary Varieties by Acreage** | **HHI Values for Proprietary Varieties by Production** |
|---|---|---|
| 2000 | 0.0376 | 0.0729 |
| 2001 | 0.0900 | 0.1474 |
| 2002 | 0.0961 | 0.1709 |
| 2003 | 0.0755 | 0.1416 |
| 2004 | 0.0898 | 0.1586 |
| 2005 | 0.0904 | 0.1425 |
| 2006 | 0.0948 | 0.1791 |
| 2007 | 0.1200 | 0.2100 |
| 2008 | 0.1533 | 0.2441 |
| 2009 | 0.1642 | 0.2593 |
| 2010 | 0.1393 | 0.1903 |
| 2011 | 0.1496 | 0.2050 |
| 2012 | 0.1149 | 0.1618 |
| 2013 | 0.2024 | 0.2882 |
| 2014 | 0.1822 | 0.2700 |

**Table 2.** *Cont.*

| Crop Year | HHI Values for Proprietary Varieties by Acreage | HHI Values for Proprietary Varieties by Production |
|:---:|:---:|:---:|
| 2015 | 0.1841 | 0.2500 |
| 2016 | 0.1832 | 0.2292 |
| 2017 | 0.2661 | 0.3170 |
| 2018 | 0.3094 | 0.3628 |
| 2019 | 0.4058 | 0.4371 |
| 2020 | 0.4927 | 0.5394 |

The U.S. proprietary hop varieties used to calculate market concentration relative to public varieties between 2000 and 2020 listed in alphabetical order: Ahtanum ™, YCR 1, Amarillo ® VGXP01, Apollo ™, Azacca ™ ADHA-483, Bravo ™, Calypso ™, Chelan, Citra ®, HBC 394, Columbus/Tomahawk®/Zeus (also known as: C/T/Z®), Ekuanot ™, HBC 366, El Dorado ®, Eureka ™, IDAHO 7™, Idaho Gem™, Jarrylo ™, ADHA-881, Loral ™, HBC 291, Millennium®, Mosaic ®, HBC 369, Pahto ™, HBC 682, Palisade ®, YCR 4, Pekko ™, ADHA-871, Sabro™, HBC 438, Simcoe ®, YCR 14, Strata™ OR 91331, Summit ™, Super Galena ™, Talus®, Warrior ™, YCR 5, Zappa® (own study based on [13]).

The HHI analysis demonstrates that the market concentration due to the increasing proportion of proprietary varieties rose from a low to a moderate concentration between 2000 and 2010. It remained in the moderate zone until 2016 when it rapidly began to increase through 2020 when a tipping point had been reached (Figures 2 and 3). Official government data documented that in 2017 proprietary varieties represented greater than 50% of U.S. hop acreage [13].

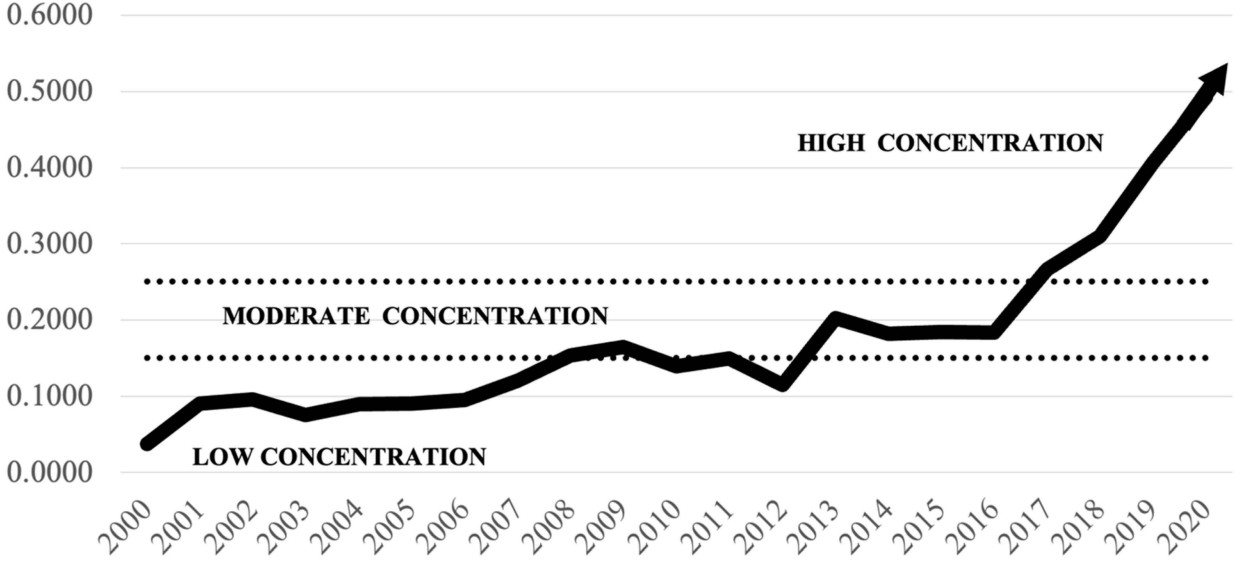

**Figure 2.** The HHI for total U.S. branded proprietary variety acreage 2000–2020 (own study based on [13]).

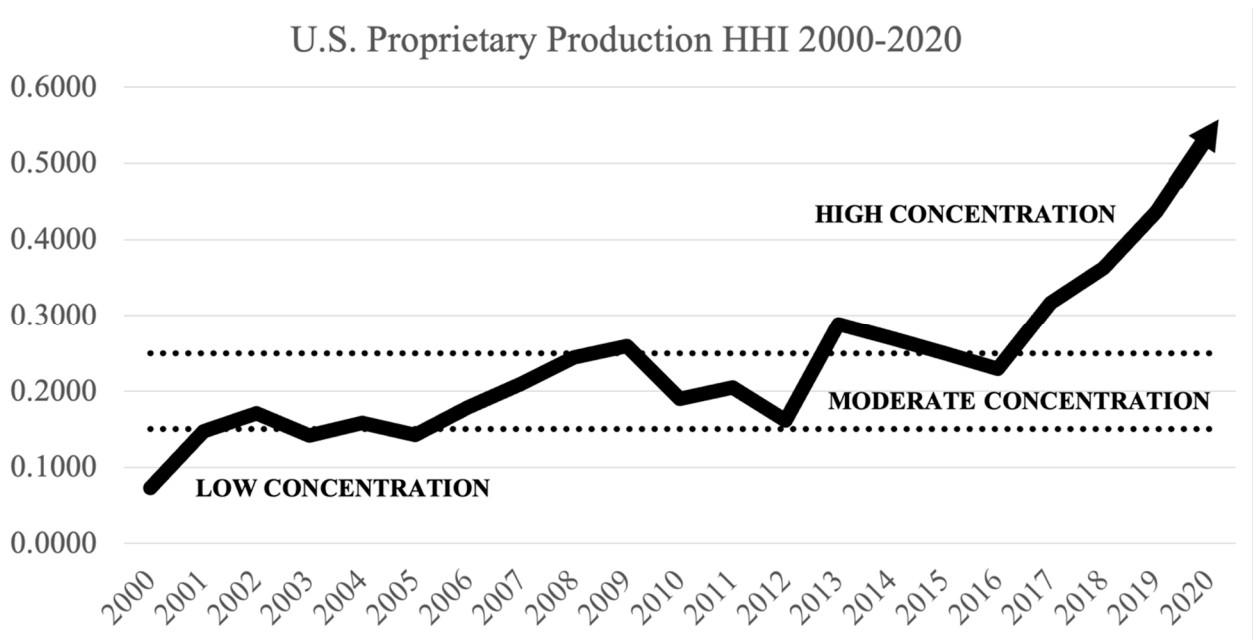

**Figure 3.** The HHI for total U.S. branded proprietary variety production 2000–2020 (own study based on [13]).

### 3.2. Prices of Hops Linked to the Intellectual Property

In the measurement of the effects on price over time, it was necessary to adjust for inflation. No appropriate Producer Price Index existed that could be applied to the U.S. hop industry. Therefore, we decided to use the Consumer Price Index (CPI) as it reflects changes in the economy and the purchasing power of the U.S. dollar over time. Vermeulen [22] suggests that U.S. producers adjust their prices as often as retailers. This suggested that the use of the CPI for adjusting prices for inflation would be an appropriate strategy.

Between 2016 and 2022, U.S. season average prices for hops as reported by the USDA remained stable at 75-year record high levels as acreage and production of proprietary varieties surpassed 50%. This suggested the existence of a tipping point facilitated by proprietary varieties that had been reached with regard to industry influence and its effect on price control. Prices when adjusted for inflation rose rapidly following 2016. The rapidly increasing HHI values post-2016 represented rapidly decreasing competitiveness. Reduced competitiveness was both a symptom and a consequence of the predominance of patented products where five entities captured 70% market share. As a result, prices increased to two standard deviations above the 75-year average of inflation-adjusted prices of $4.56 per pound (Figure 4). The parameter of long-term season average prices of hops during that time was chosen because it demonstrated lower price variability than the previous 36-year period for which data was available [13], a period that included World War I, the Great Depression, U.S. Prohibition, and World War II.

Rhoades (1995) concluded that the results of such analyses can yield useful insights into industry behavior. Concentration and the degree of competitiveness within an industry can impact price. MacAvoy [11] identified a general hypothesis regarding changes in market concentration and price-cost margins used to determine the direction of competitiveness.

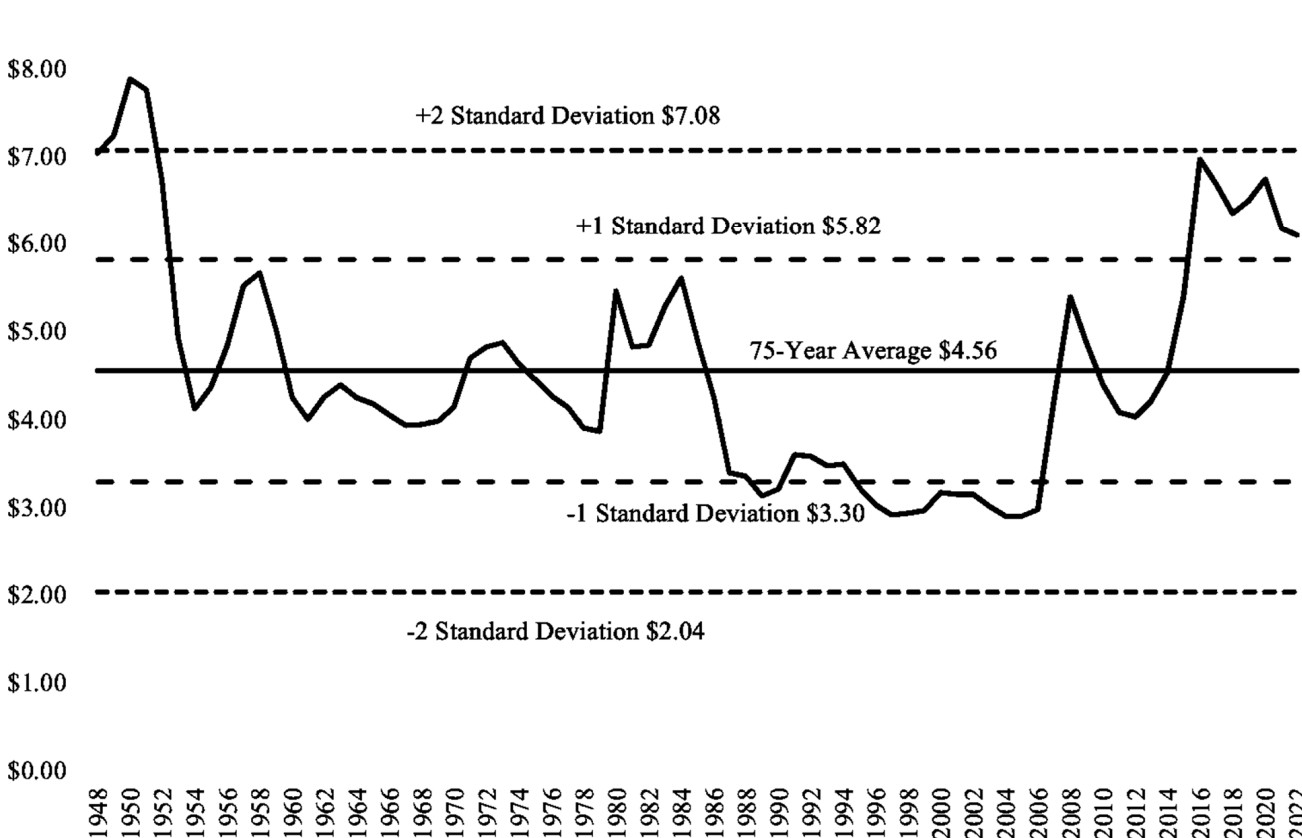

**Figure 4.** U.S. Season Average Prices 1948–2022 Adjusted for Inflation Using CPI (own study based on [16,23,24]).

Industry concentration reduced price competition as licensing agreements centralized decisions regarding production, sales, and marketing. Standardized quality by controlling harvest timing by IP owners was another result [25,26]. Patents grant the inventor control over the production, sales, and licensing of their invention without government involvement or oversight. Quantity or volume regulation and producer allotments of hops through Federal Marketing Orders have previously led to monopolistic policies [27].

*3.3. Discussion*

The Herfindahl-Hirschman Index calculations revealed that increases in proprietary variety acreage resulted in increased concentrations of power within the industry. This resulted in reduced competitiveness. Reduced competition in an oligopoly such as that which existed in the hop industry between 2016 and 2022 resulted in inflation-adjusted prices remaining stable at levels between one and two standard deviations higher than the 75-year average of U.S. season average prices, which was $4.56 per pound of hops i.e., $2.06 per kg of hops despite the existence of a growing surplus of hop inventory [13]. This was contrary to the normal tendency for prices to revert to long-term mean values or lower within 2–3 years following a price spike.

Reduced competitiveness within the hop industry during the period 2000–2020, enabled season average prices to remain at elevated levels for a prolonged period as they did between 2016 and 2020. The intrinsic homogeneous traits of branded proprietary varieties of hops such as oil production, which would typically result in symmetrical marginal costs, are overshadowed by extrinsic heterogeneous characteristics such as the perceived value of a brand and the urgency created by artificial scarcity. These characteristics create the

perception of additional value for which the brewing industry has been prepared to pay handsomely [28]. The premium price and royalties warranted by proprietary varieties can be considered a deadweight loss ultimately born by the beer consumer.

We expanded the variety-specific acreage market share calculations to group those varieties that share common ownership to get a better picture of the influence of the five largest variety development companies. One company, the Hop Breeding Company, had a much greater share than the rest. Common ownership between the entities that create branded proprietary varieties, individual hop farms and hop merchant firms further increased market concentration. The individuals who own the entities that create proprietary varieties have created a competitive advantage for the merchant companies and farms in which they share a financial interest. We concluded that branded proprietary varieties, when their ownership is concentrated in few hands, reduced competition within the market, encouraged market segmentation and created opportunities for potential anti-competitive behavior.

According to data available between 2009 and 2020 from IHGC economic reports and the Hop Growers of America Statistical Packets, the farmgate value for contracted American hops was $2.88 billion greater than German growers. That does not represent the added value that processing, packaging, and reselling add to the price paid by brewers and beer consumers. During this same period, the USDA reported that proprietary variety acreage increased from 40.52 percent to 70.19 percent in the PNW [13].

## 4. Conclusions

The Herfindahl-Hirschman Index calculations offered a glimpse of changes in proprietary variety market share and the impact these changes have had upon market concentration and competitiveness within the U.S. hop industry between 2000 and 2022. During that short time, the industry went from one dominated by publicly available varieties to one managed product controlled by a duopoly.

The most relevant of the consequences of increased market concentration of reduced competitiveness was the greater ability of proprietary variety owners to manage the perception of scarcity of supply for their proprietary products on the market. Artificial scarcity created fear among brewers, which led to them signing long-term contracts at high prices (terms dictated by the farmer/merchant entities). Through their efforts, they could reduce or eliminate surplus inventory thereby enabling sustained premium prices indefinitely. A return to a system based on free market competition rather than one controlled by oligarchs would return market-based pricing and eliminate prices set at artificially high levels.

Additional data now suggest a surplus of contracted proprietary varieties developed and grew between 2016 and 2022. Artificially high prices were sustained since free market forces were not allowed to act. In the face of oversupply, these higher prices artificially increased the cost of production not only for American brewers but for brewers around the globe. The effect of proprietary varieties on the DSR remains to be seen as it is underway in 2023. It appears at the time of this writing the additional opacity created by private management of approximately 70 percent of the U.S. crop delayed the initial signaling period for the DSR to begin.

The effects of such supply management efforts affect not only proprietary U.S. varieties but public varieties in the U.S., too. The relationship between hop varieties (i.e., hop varieties may be substituted with other varieties) extends the effects of proprietary variety management upon farmers in countries where they are not produced. Additional research regarding the complementary relationship between the U.S. and German production regions is recommended to understand price movements, the disparity of pricing, and perceived value. The reduction in competitiveness within the U.S. industry this research provides is an important step in furthering the understanding of hop market dynamics and the interrelatedness of world markets. Further research might include an examination of the methods owners of proprietary varieties may use to cooperate with other related

entities to alter supply to determine where the border exists for anti-trust violations. Further investigation into acreage management strategies is warranted.

**Author Contributions:** Conceptualization, methodology and improving the manuscript, D.M. and M.P.; results interpretation, writing and editing, D.M.; supervision, project administration, M.P. All authors have read and agreed to the published version of the manuscript.

**Funding:** This research was funded by the Slovenian Research Agency (ARRS) and the University of Maribor (FKBV).

**Data Availability Statement:** Not applicable.

**Conflicts of Interest:** The authors declare no conflict of interest.

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
