# Peer review of "Proprietary Varieties’ Influence on Economics and Competitiveness in Land Use within the Hop Industry"

_land, doi:10.3390/land12030598_

Round 1
Reviewer 1 Report
This is an excellent piece of work. The manuscript provides a detailed analysis of intellectual property, competitiveness and market concentration dynamics for the Hop Industry. The manuscript can be published in the current form.
Author Response
Dear Fellow,
We do appreciate your reviewer's opinion.
The Authors
Reviewer 2 Report
see attached.

Author Response
Dear Reviewer,
we do appreciate your opinion that enabled us to improve the paper proposal within the Ph.D. research work. We revised the manuscript and believe to address the reasonable concerns.
The updated text in the chapters 1, 2 and 4 (pages 2, 3 and 10) ameliorate the paper’s background, methodological approach and conclusion with research limitations. The Journal citation standards were respected as well as the paragraph formatting. Furthermore, the additional seven relevant scientific references enriched the research content.
However, in the proposal we could not follow those of the recommendations, which might have led to parts of the paper being based simply on industry heresy and speculation - which we wanted to avoid.
Thus, we ask for your consideration in publishing the article in the journal Land.
The Authors
Reviewer 3 Report
The manuscript entitled "Proprietary Varieties Influence on Economics and Competitiveness In Land Use Within a Hop Industry" (land-2229401) was aimed at assessing the level of competitiveness in the case of the U.S. hop industry through the lens of the Herfindahl-Hirschman Index (HHI).
To improve the quality of the manuscript land-2229401, I propose the following:
(1) The abstract should be adapted according to the journal's recommendation, especially according to these points: the abstract should include information on: the background – a broad and brief presentation of the general context of the analyzed issue (which is now lacking); and the conclusions – indicate the main implications (the main findings were presented, but managerial implications are lacking).
(2) Considering the title of the manuscript, the authors should emphasize the land use impact on the "highly concentrated" U.S. hop industry and propose recommendations, and improvement measures at all value chain levels.
(3) More literature background is required as far as the concept of competitiveness is concerned, especially concerning land competitiveness. More reflexivity is expected. Please ground the use of the Herfindahl-Hirschman Index better in the literature from the competitiveness perspective and highlight the importance of concentration analysis in this regard.
(4) If the authors consider it suitable, it would be interesting to tap into price volatility and its impact on capitalizing on building high levels of competitiveness in the hop industry.
(5) The current research limitations should be detailed in relation to future research avenues or perspectives.
(6) Please pay more attention to formatting and referencing style (kindly ask you to check the journal's template).
Author Response
Dear Fellow reviewer,
we do appreciate your reviewer's opinion that helped us to improve the paper proposal within our PhD research work. We revised the manuscript and believe to address the reasonable concerns.
We consider the updated abstract to embrace the journal’s main points of information. In addition, texts in the chapters 1, 2 and 4 (pages 2, 3 and 10) ameliorate paper’s background, methodological approach and conclusion with research limitations. The Journal citation standards were respected as well as the paragraph formatting. Furthermore, the 7 additional relevant scientific references enriched the research content.
Thus, we ask for your consideration in publishing the article in the journal Land.
The Authors
Reviewer 4 Report
I would suggest the authors make a few formal edits:
· Non-uniformity of the citation standard. According to the officiating Land magazine template, the source number is given in square brackets after the cited text).
· Inconsistency in paragraph formatting (line spacing).
· I would recommend writing the formula for HHI calculation in the equation editor.
In terms of content, it would be appropriate:
· To add a chapter „Research limitations“.
· To add other international sources in Discussion (etc. „Pavlovic, M. PRODUCTION CHARACTER OF THE EU HOP INDUSTRY. BULGARIAN JOURNAL OF AGRICULTURAL SCIENCE, 2012, vol. 18, No. 2, pp. 233-239 “, „ Sredl, K., Prasilova, M., Svoboda, R., Severova, L. Hop production in the Czech Republic and its international aspects. Heliyon, 2020, vol. 6, No. 7“ or citation from Land journal).
Otherwise, I recommend the paper for publication.
Author Response
Dear Fellow,
We do appreciate your reviewer's opinion that helped us to improve the paper proposal within the Ph.D. research work.
Journal citation standards were properly respected as well as the paragraph formatting. In addition, the 7 additional relevant references enriched the research content. Furthermore, research limitations are included within the chapter Conclusions.
Respectfully,
The Authors